# The Cannabinoids, CBDA and THCA, Rescue Memory Deficits and Reduce Amyloid-Beta and Tau Pathology in an Alzheimer’s Disease-like Mouse Model

**DOI:** 10.3390/ijms24076827

**Published:** 2023-04-06

**Authors:** Juyong Kim, Pilju Choi, Young-Tae Park, Taejung Kim, Jungyeob Ham, Jin-Chul Kim

**Affiliations:** 1Department of Agricultural Biotechnology, Seoul National University, Seoul 08826, Republic of Korea; 2Natural Product Research Center, Korea Institute of Science and Technology (KIST), Gangneung 25451, Republic of Korea; 3Division of Bio-Medical Science & Technology, KIST School, University of Science and Technology, Seoul 02792, Republic of Korea; 4NeoCannBio Co., Ltd., Gangneung 02792, Republic of Korea

**Keywords:** Alzheimer’s disease, apoptosis, CBDA, THCA, cannabinoid, calcium

## Abstract

Most studies related to hemp are focused on Cannabidiol (CBD) and Tetrahydrocannabinol (THC); however, up to 120 types of phytocannabinoids are present in hemp. Hemp leaves contain large amounts of Cannabidiolic acid (CBDA) and Tetrahydrocannabinolic acid (THCA), which are acidic variants of CBD and THC and account for the largest proportion of CBDA. In recent studies, CBDA exhibited anti-hyperalgesia and anti-inflammatory effects. THCA also showed anti-inflammatory and neuroprotective effects that may be beneficial for treating neurodegenerative diseases. CBDA and THCA can penetrate the blood–brain barrier (BBB) and affect the central nervous system. The purpose of this study was to determine whether CBDA and THCA ameliorate Alzheimer’s disease (AD)-like features in vitro and in vivo. The effect of CBDA and THCA was evaluated in the Aβ_1–42_-treated mouse model. We observed that Aβ_1–42_-treated mice had more hippocampal Aβ and p-tau levels, pathological markers of AD, and loss of cognitive function compared with PBS-treated mice. However, CBDA- and THCA-treated mice showed decreased hippocampal Aβ and p-tau and superior cognitive function compared with Aβ_1–42_-treated mice. In addition, CBDA and THCA lowered Aβ and p-tau levels, alleviated calcium dyshomeostasis, and exhibited neuroprotective effects in primary neurons. Our results suggest that CBDA and THCA have anti-AD effects and mitigate memory loss and resilience to increased hippocampal Ca^2+^, Aβ, and p-tau levels. Together, CBDA and THCA may be useful therapeutic agents for treating AD.

## 1. Introduction

Alzheimer’s disease (AD) is an age-related neurodegenerative disease accompanied by memory and cognitive deficits [1]. It has been more than 100 years since AD was first reported. However, there is currently no treatment for AD. The causes of AD are unclear, although several hypotheses include the amyloid-beta hypothesis and the highly phosphorylated tau hypothesis [2,3]. The accumulation of amyloid-beta and hyper-phosphorylated tau cause neuronal cell death, synaptic collapse, and neuro-inflammation, which are hallmark symptoms of AD. This results in a decrease in memory and cognitive function, leading to severe dementia [4].

The production of amyloid-beta and hyper-phosphorylated tau is associated with a high intracellular calcium concentration, which activates enzyme-related amyloid-beta production and promotes tau phosphorylation [5]. For demented patients, the calcium concentration in nerve cells is higher compared with that of ordinary people, and memory cannot be created [6]. Calcium acts as a signal transmitter in cells and is involved in various signaling pathways, such as enzyme activation, protein expression, gene transcription, and programmed cell death [7,8,9,10]. Calcium also plays important roles in nerve cells’ synaptic plasticity, memory formation, and neurogenesis [11,12]. Calcium dyshomeostasis affects various signaling pathways in the nerve cells. Moreover, these signals cause damage to nerve cells and eventually develop into severe AD [13,14].

As described above, Ca^2+^ performs an essential and fundamental function in nerve cell functioning. Cannabinoid receptors are involved in calcium homeostasis through various mechanisms. In particular, the activation of CB1 and CB2 inhibits N-methyl-D-aspartate (NMDA) receptors and lowers calcium concentrations, showing neuroprotective effects [15,16]. It is also known that the agonists of CB inhibit various voltage-gated calcium channels (VGCCs), including N- and T-type calcium channels, thereby lowering the calcium concentration [17,18].

Cannabidiolic acid (CBDA) and Tetrahydrocannabinolic acid (THCA) are known as the agonists of CB [19,20] and acidic variants of Cannabidiol (CBD) and Tetrahydrocannabinol (THC), respectively [21]. There are up to 120 phytocannabinoids present in hemp, and CBDA and THCA account for a large proportion [22]. CBDA exhibits anti-hyperalgesia, anti-inflammatory, and anti-nausea effects [23,24]. It also reduces seizure, anxiety, and depression in a mouse model [25,26]. THCA has anti-inflammatory, neuroprotective, anti-convulsant, and anti-seizure effects [27,28,29]. CBDA and THCA inhibit T-type calcium channels [30]. In a pharmacokinetics study, CBDA and THCA exhibited a higher Cmax in serum compared with CBD and THC [31]. CBDA and THCA function directly in the brain because of their ability to cross the blood–brain barrier (BBB) [32]. Although no clinical trials have been reported thus far, CBDA and THCA may have increased efficacy and bioavailability compared with CBD and THC.

In this study, we hypothesized that the cannabinoids, CBDA and THCA, ameliorate AD-like features by modulating Ca^2+^ levels, hippocampal pathology, and cognitive decline. We determined the effects of CBDA and THCA on the pathogenesis of AD in an AD-like mouse model (Aβ_1–42_-treated mice) by unilateral injection of Aβ_1–42_ into the hippocampus [33]. In addition, intracellular Ca^2+^ levels, Aβ, tau, and p-tau production were examined in primary neuronal cell cultures during AD-related Aβ pathology development.

## 2. Results

### 2.1. CBDA and THCA Treatment Decreases Cell Death and Ca^2+^ Levels in Primary Cultures of Cortical Neurons

Cortical neurons from ICR mice were cultured for 6 days. Primary cortical neurons were treated with Aβ_1–42_ (5 μM) and/or CBDA (3 and 6 μM) or THCA (3, 6, and 12 μM) for 24 h. Neuronal cell death was markedly increased in primary neurons treated with Aβ_1–42_ (5 μM) by 70% (*p < 0.001*), whereas CBDA (3 μM (75% (*p < 0.001*)) and 6 μM (78% (*p = 0.009*)) and THCA (6 (79% (*p = 0.004*)) and 12 μM (79% (*p < 0.001*)) significantly suppressed neuronal cell death (Figure 1A,B). In addition, we measured Ca^2+^ by staining with Fluo-4 AM to determine the effect of CBDA and THCA on intracellular Ca^2+^ levels. The fluorescence intensity of Ca^2+^ was significantly increased in Aβ_1–42_-treated neurons compared with PBS-treated neurons (100% (*p < 0.001*)); however, this increase was significantly ameliorated by 6 μM CBDA (32% (*p = 0.005*)) and 12 μM THCA (51% (*p = 0.010*)) treatment (Figure 1C,D).

### 2.2. CBDA and THCA Treatment Decreases Aβ and p-Tau Levels in Primary Neurons

To determine the effect of CBDA and THCA on Aβ aggregation and p-tau in primary cortical neurons, Western blot analysis was done to measure Aβ and p-tau (AT8) expression levels. The levels of APP (228% (*p = 0.001*)), polymeric Aβ (188% (*p = 0.003*)), and oligomeric Aβ (261% (*p = 0.005*)) were significantly increased in neurons treated with Aβ_1–42_ compared with neurons treated with PBS, whereas this increase was reversed following treatment with CBDA (APP; 152% (*p = 0.024*), polymeric Aβ; 82% (*p = 0.001*), oligomeric Aβ; 89% (*p = 0.004*)) and THCA (APP; 141% (*p = 0.013*), polymeric Aβ; 90% (*p = 0.001*), oligomeric Aβ; 89% (*p = 0.004*)) (Figure 2A–D). In addition, the level of p-tau (AT8) in neurons treated with Aβ_1–42_ was also significantly higher compared with that in PBS-treated neurons (189% (*p = 0.002*)), which was significantly mitigated by CBDA (114% (*p = 0.005*)) and THCA (138% (*p = 0.003*)) treatment (Figure 2E–G).

### 2.3. CBDA and THCA Treatment Ameliorates Learning and Memory Loss in Aβ_1–42_-Treated Mice

To determine the effect of CBDA and THCA on the pathogenesis of AD, the hippocampus of the mice was unilaterally infused with Aβ_1–42_ (3 μg/mouse) or PBS. Two days after injection, CBDA (6 μmol/mouse) or THCA (12 μmol/mouse) was similarly injected into the hippocampus of Aβ_1–42_-treated mice to determine the effect of CBDA and THCA on learning and memory. We conducted Morris water maze and object location tests to evaluate spatial learning ability and novel object recognition tests to assess the ability to recognize new objects (Figure 3). The experimental schedule for the behavioral tests is summarized in Figure 3A. Two-way ANOVA analysis of mean escape latency (i.e., the time required to locate the escape platform) in the Morris water maze test revealed statistically significant differences between the three groups (interaction (*p = 0.047*), Treatment (*p < 0.0001*), Time (*p < 0.0001*)). Aβ_1–42_-treated mice learned the location of the submerged platform more slowly compared with PBS-treated mice during training sessions and showed less improvement throughout training. However, mice treated with CBDA or THCA following Aβ_1–42_ treatment performed better than those treated with Aβ_1–42_ alone (Figure 3B). On day 5 of the probe test, Aβ_1–42_-treated mice remained in the target quadrant (*p < 0.001*) and platform area (*p = 0.035*) for a significantly shorter time compared with mice treated with PBS. For Aβ_1–42_-treated mice, CBDA or THCA treatment resulted in a longer time in the target quadrant (CBDA; *p = 0.008*, THCA; *p = 0.030*) and platform area (CBDA; *p = 0.059*, THCA; *p = 0.114*) (Figure 3C,D). The number of times crossing the platform area was significantly reduced in Aβ_1–42_-treated mice compared with the PBS-treated mice (*p = 0.025*). CBDA (*p = 0.026*) or THCA (*p = 0.044*) treatment resulted in an increase in the number of crossings in Aβ_1–42_-treated mice (Figure 3E). During the novel object phase, mice treated with Aβ_1–42_ + CBDA (*p < 0.001*) or THCA (*p < 0.001*) spent more time exploring the novel object and exhibited significantly higher discrimination ratios compared with Aβ_1–42_-treated mice (Figure 3F). In the object location test, mice treated with Aβ_1–42_ + CBDA (*p < 0.001*) or THCA (*p < 0.001*) also spent more time examining the displaced object. They exhibited significantly higher discrimination ratios compared with Aβ_1–42_-treated mice (Figure 3G).

### 2.4. CBDA and THCA Treatment Decreases Aβ and p-Tau Levels in the Hippocampus of Aβ_1–42_-Treated Mice

In the acute AD-like mouse model injected with amyloid beta, amyloid beta aggregation, and tau pathology, which are representative pathological markers of AD, occur [34]. So, to determine the effect of CBDA and THCA on hippocampal Aβ aggregation and p-tau in Aβ_1–42_-treated mice, hippocampal tissue was collected from five mice from each group 19 days after the initial Aβ_1–42_ infusion. We conducted a Western blot analysis to measure hippocampal Aβ and p-tau (AT8) expression levels. The levels of hippocampal APP (479% (*p < 0.001*)), polymeric Aβ (202% (*p = 0.001*)), and oligomeric Aβ (198% (*p = 0.004*)) were significantly increased in Aβ_1–42_-treated mice compared with that in PBS-treated mice, and these increases were reversed by CBDA (APP; 140% (*p < 0.001*), polymeric Aβ; 66% (*p < 0.001*), oligomeric Aβ; 101% (*p = 0.007*)) and THCA (APP; 131% (*p < 0.001)*, polymeric Aβ; 81% (*p < 0.001*), oligomeric Aβ; 103% (*p = 0.008*)) treatment (Figure 4A–D). The levels of hippocampal p-tau (AT8) in Aβ_1–42_-treated mice were also significantly higher compared with that in PBS-treated mice (160% (*p = 0.002*)), which were significantly mitigated by CBDA (116% (*p = 0.032*)) and THCA (105% (*p = 0.008*)) treatment (Figure 4E–G).

### 2.5. CBDA and THCA Treatment Modulates BDNF/CREB Signaling Pathway in the Hippocampus of Aβ_1–42_-Treated Mice

Brain-derived neurotrophic factor (BDNF) and its receptor protein p-TrkB are major factors in learning, memory formation, synaptic plasticity, and neuroprotection [35,36]. Because BDNF is regulated through CREB phosphorylation, CREB plays a very important role in memory formation [37]. The phosphorylation of CREB is regulated by changes in calcium concentration; however, when calcium concentration is overloaded, p-CREB is dephosphorylated and does not function [38]. Therefore, we determined the effect of CBDA and THCA on BDNF, p-TrkB, and p-CREB levels in the hippocampus of the AD-like mouse model by Western blot analysis. Aβ_1–42_-treated mice had significantly decreased hippocampal BDNF (31% (*p < 0.001*)), p-TrkB (65% (*p = 0.001*)), and p-CREB (55% (*p = 0.011*)) levels compared with PBS-treated mice; however, hippocampal BDNF, p-TrkB, and p-CREB levels were significantly increased by CBDA (BNDF 83% (*p = 0.001*), p-TrkB 90% (*p = 0.001*), p-CREB 95% (*p = 0.023*)) and THCA (BNDF 119% (*p < 0.001*), p-TrkB 113% (*p = 0.008*), p-CREB; 92% (*p = 0.012*)) treatment (Figure 5A–D).

## 3. Discussion

The typical symptoms of Alzheimer’s disease are cognitive and memory impairment [39]. Representative pathological markers of AD are considered increased Aβ and p-tau, which result in neuronal cell death [40]. The cause of Alzheimer’s disease is unclear, but studies have indicated that calcium dyshomeostasis is a major contributor [41]. In neurons, calcium activates and deactivates various signaling pathways. When calcium homeostasis is dysregulated, various signaling events collapse, causing impairment of learning and memory [42]. Patients with AD have higher calcium concentrations compared with ordinary people. As a result of increased calcium concentrations, long-term potentiation cannot occur, and memory cannot form [6,43]. In addition, in AD, the expression level of BDNF and the activity of the BDNF/CREB signaling pathway are decreased [44]. BDNF is attractive as a potential evaluation marker for the efficacy of AD treatments [45,46]. Several experiments have shown that BDNF overexpression or its injection into AD animal models demonstrated therapeutic efficacy for AD [47,48]. The BDNF/CREB signaling pathway is regulated by Ca^2+^ [49]. p-CREB is a transcription factor involved in BDNF expression and is activated by Ca^2+^ influx. However, excessive Ca^2+^ influx causes dephosphorylation of p-CREB and reduction of BDNF expression [38]. 

The FDA approved Memantine, an antagonist of the N-methyl-D-aspartate (NMDA) calcium receptor, as a drug for treating AD [50]. Memantine lowers intracellular calcium concentrations by inhibiting the NMDA calcium receptor [51]. A decrease in calcium concentration following Memantine treatment results in neuroprotection, learning, and cognitive function, exhibits inhibitory effects on Aβ and p-tau production and activates the BDNF/CREB signaling pathway [52,53].

Although hemp is classified as a narcotic with many restrictions, studies have demonstrated its efficacy in treating various diseases [54]. There are various species of hemp, but only the Cannabis sativa. L strain, which is categorized according to THC and CBD content, is used [55]. Moreover, its receptor, the cannabinoid receptor, performs various functions in the nervous system. Cannabinoid receptors are expressed in astrocytes, microglia, and neurons [56,57]. Furthermore, cannabinoid receptors are involved in the survival of nerve cells, synaptic plasticity, and the development of dendrites [58,59,60]. Cannabinoid receptors are expressed not only in the peripheral nervous system but also in the central nervous system [61]. Thus, it affects various neurodegenerative diseases, including Parkinson’s disease and Alzheimer’s disease [62,63]. Therefore, various studies have been conducted on the function of cannabinoid receptors in the central nervous system. In particular, many studies have been conducted on neurodegenerative diseases using CBD and THC, representative agonists of cannabinoid receptors [64,65]. In addition, CBD and THC have potential therapeutic efficacy in clinical trials for Parkinson’s disease [66,67]. Accordingly, the development of drugs targeting cannabinoid receptors is progressing [68].

Several studies have shown that the activation of cannabinoids has various advantages, such as neuroprotective effects [69]. More than 120 types of phytocannabinoids exist in hemp, but most studies have focused on CBD and THC [70]. However, CBDA and THCA, the acidic variants of CBD and THC, account for a large proportion of hemp leaves [22]. In a pharmacokinetics study, CBDA and THCA were present in the serum at higher concentrations compared with CBD and THC [31]. CBDA can directly affect the brain because of its ability to penetrate the BBB [32]. CBDA showed anti-convulsant, anti-hyperalgesia, anti-inflammation, anti-nausea, anti-anxiety, and anti-seizure effects in animal models [23,25,71]. In addition, THCA has BBB penetration ability; it exhibits anti-convulsant, anti-inflammation, and anti-nausea effects in animal models as well as neuroprotection by inhibiting various inflammatory cytokines in cell models [29,32,72,73]. Furthermore, CBDA and THCA inhibit calcium influx by acting as antagonists in T-type calcium channels [30]. These characteristics offer a novel approach to treating AD.

Increased intracellular calcium concentration by Aβ reduces BDNF levels [74]. In the present study, we hypothesized that CBDA and THCA normalize calcium concentration to increase BDNF levels and inhibit Aβ and p-tau production, thereby inhibiting neuronal apoptosis and improving cognitive function. We found that CBDA and THCA modulate Ca^2+^ influx, exhibit neuroprotective effects, and reduce Aβ and p-tau production against Aβ_1–42_ in primary neurons. In addition, CBDA and THCA decreased the production of Aβ and p-tau, promoted CREB phosphorylation, a transcription factor of BDNF, and consequently increased the expression of BDNF and its receptor, p-TrkB, in the hippocampus of Aβ_1–42_-treated mice. CBDA and THCA rescued object and spatial cognitive function and memory deficits in Aβ_1–42_-treated mice. Overall, these results suggest that CBDA and THCA ameliorate AD-like features by modulating Ca^2+^ homeostasis, which is fundamental to neuronal viability and function.

## 4. Materials and Methods

### 4.1. Animals

Female ICR mice (8 weeks) were purchased from the Koatech company (Pyeongtaek, Republic of Korea). The mice were housed in the animal care facility (temperature 22 ± 2 °C; humidity 40–60%, and a 12 h light/dark cycle) at the Korea Institute of Science and Technology (KIST). The mice were provided food and water ad libitum.

### 4.2. Drugs and Reagents

Chongsam (Korean hemp, *Cannabis sativa* L.) was collected from the Association (Andong city, Gyeongsangbuk-do, Republic of Korea) in accordance with assignment/transfer approval process (approval No. 1564) stipulated by the Korean Ministry of Food and Drug Safety and the Seoul Regional Food and Drug Administration. Chongsam leaves were harvested in July 2019, naturally dried, and finely cut, and 10 g was extracted twice with ethanol (200 mL) at room temperature and filtered. The ethanolic extract (1.64 g) was suspended in water and successively partitioned with normal hexane, which yielded 720 mg of residue. Silica open column chromatography (Merck, 230–400 mesh, 2.0 × 10.0 cm ID) was carried out using hexane: ethyl acetate (F1–10:0, F2–25:1, F3–10:1, and F4–0:10; each 200 mL) stepwise gradient. The F2 (187 mg) fraction was subjected to preparative HPLC (Phenomenex Luna C18 column; 250 × 21.2 mm, 10 μm) and eluted using a water (A) and MeCN (B) gradient system (70–85% MeCN over 60 min) with a 10 mL/min flow rate and UV detection at 220 nm to yield four subfractions (a–d). Further purification of each subfraction was done using semi-preparative HPLC (Phenomenex Luna C18 (2); 250 × 10 mm, 5 μm) with 70 to 85% MeCN as an eluant at 4 mL/min flow rate to yield pure THCA (17.0 mg). Fraction F3 (35 mg) was subjected to preparative HPLC (Phenomenex Luna C18 column; 250 × 21.2 mm, 10 μm) and gradient eluted with water (A) and MeCN (B) at 65 to 80% MeCN over 60 min at 10 mL/min flow rate and a 220 nm UV detector to yield one subfraction (e). This fraction was subjected to semi-preparative HPLC (Phenomenex Luna C18 (2); 250 × 10 mm, 5 μm) using a 65 to 80% MeCN gradient at a flow rate of 4 mL/min to yield pure CBDA (7.9 mg).

Lyophilized Aβ_1–42_ (1 mg) was purchased from Cayman Chemical (Ann Arbor, MI, USA). For the treatment of primary neurons, Aβ_1–42_ was prepared in 400 μL DMSO and incubated for 1 h at room temperature (RT) with rotation. Oligo-Aβ_1–42_ was prepared by diluting Aβ_1–42_ with neurobasal medium (Gibco, Carlsbad, CA, USA) at a concentration of 100 μM. The oligo-Aβ_1–42_ was incubated for 24 h at 4 °C with rotation. For stereotaxic surgery, Aβ_1–42_ was prepared in 20 μL DMSO and incubated for 1 h at RT with rotation. Oligo-Aβ_1–42_ was prepared by diluting Aβ_1–42_ with PBS at a concentration of 1 μg/μL and then incubated for 24 h at 4 °C with rotation.

### 4.3. Primary Neuronal Culture

As previously described, the cerebral cortical tissue was dissected from day 15 embryonic ICR mice [75]. Cells were isolated by digestion with 0.05% trypsin and re-suspended in minimal essential medium containing 10% heat-inactivated horse serum, 10% fetal bovine serum, 2 mM glutamine, 100 units/mL penicillin, and 100 μg/mL streptomycin. The isolated cortical neurons were allowed to adhere to 0.2 mg/mL poly-D-lysine-coated culture dishes for 45 min and cultured in neurobasal medium supplemented with B27 (Gibco, Waltham, MA, USA), 1 mM glutamine, 100 units/mL penicillin, and 100 μg/mL streptomycin. Cultures at 6 days were treated with Aβ_1–42_ and/or CBDA or THCA for 24 h, and the neurons were harvested for Western blot analysis to measure APP/Aβ, tau, and p-tau levels.

### 4.4. Cell Viability

Primary neurons (5 × 10^5^ cells/well) were seeded into 96-well plates for 6 days. Aβ_1–42_ and/or CBDA or THCA was added to the cells for 24 h. MTT solution (Invitrogen, Carlsbad, CA, USA) was added to the medium and incubated for 2 h. Cytotoxicity was measured using a microplate spectrophotometer (Bio-Tek Power Wave XS, Winooski, VT, USA) at 490 nm.

### 4.5. Fluorescence Ca^2+^ Imaging

Primary neurons, 2 × 10^6^ cells per well, were cultured in 6-well for 6 days to evaluate intracellular Ca^2+^. Cultures at 6 days were treated with Aβ_1–42_ and/or CBDA or THCA for 24 h, washed with PBS, and loaded with 10 μM Ca^2+^ indicator Fluo-4 AM (Invitrogen, Carlsbad, CA, USA) for 1 h. The stained sections were cleaned with PBS, a coverslip with Prolong Gold Antifide Reagent containing DAPI nuclear stain was added (Invitrogen, Carlsbad, CA, USA), and the samples were examined using a microscope (Carl Zeiss, Oberkochen, Germany). Regions of interest that were 422,500 μm^2^ were randomly selected from each well and measured 3-4 areas per well. The Image J analysis program measured the entire fluo-4 AM fluorescence signal intensity (National Institute of Health, Bethesda, MD, USA).

### 4.6. Intrahippocampal Stereotaxic Injection of Aβ_1–42_

Eight-week-old mice were acclimatized to laboratory conditions for one week and divided into four groups: PBS-treated mice (PBS + PBS), Aβ_1–42_-treated mice [Aβ_1–42_ (1 μg/μL, 3 μL/mouse) + PBS], CBDA-treated mice [Aβ_1–42_ + CBDA (6 μM, 3 μL/mouse)], and THCA-treated mice [Aβ_1–42_ + THCA (12 μM, 3 μL/mouse)]. After anesthetizing with an intraperitoneal injection of avertin (250 mg/kg), the mice were immobilized on a stereotaxic instrument. Aβ_1–42_ (days 0 and 1), CBDA or THCA (days 3 and 4), or PBS (days 0, 1, 3, and 4) was administered (3 μL/15 min/mouse) to the hippocampus (coordinates from bregma: mediolateral (ML) = 1.30 mm, anteroposterior (AP) = −2.00 mm, and dorsoventral = −2.20 mm) gradually using a Hamilton syringe.

### 4.7. Morris Water Maze Test

A modification of the water maze procedure described by Morris was used to examine cognitive function [76]. A circular tank (diameter 90 cm, height 50 cm; 22 ± 2 °C water temperature) was used for the test. The tank consisted of four quadrants filled with water. An escape platform (6 cm diameter and 29 cm height) was submerged 1 cm below the water surface at the center of one of the four quadrants. Each mouse was trained for 4 days to learn and memorize visual cues placed outside the tank, which indicated platform location. The swimming paths used by each mouse were recorded with a camera connected to a video recorder and path tracking software XT (EthoVision; Noldus Information Technology, Wageningen, The Netherlands). Four trials were performed each day during the 4-day training period. During each trial, each mouse was allowed 60 s to find the hidden platform and another 30 s to stay on the platform. If the mouse was unable to find the platform within 60 s, it was guided to it and allowed to remain for 30 s. The mean time (mean escape latency) that each mouse took to find the platform was recorded. The probe test was conducted after 4 days in the same manner without the platform. Each mouse was allowed 60 s to move freely and was recorded. The video was analyzed using tracking software (EthoVision; Noldus Information Technology, Wageningen) to count the time spent in the target quadrant area and platform areas and the number of crossovers.

### 4.8. The Novel Object Recognition Test

We used a modified novel object recognition test, which incorporates the natural tendency of a mouse to explore novel stimuli [77]. During a habituation session performed 2 days before testing, the mice were allowed to explore (for 10 min) a test environment consisting of an empty opaque, custom-made Plexiglas box (35 cm × 45 cm × 25 cm). The sample object phase was introduced 24 h later. Two identical white circular cylinders (the sample objects) were placed on the facing edge in the test environment, and the mice were given access to the objects for 10 min. After 24 h (the novel object phase), one of the sample objects in the test environment was replaced with a similar-sized novel object (a colored miniature animal), and the mice were given 5 min of contact with this new arrangement. The time that the animal’s nose was <1 cm from an object was considered the time it navigated the object. The amount of time that the mouse stood on the object was excluded. The discrimination ratio was the time used to navigate the novel object over the time used to navigate both objects.

### 4.9. Object Location Test

The object location test was conducted in the same manner as described for the novel object recognition test. On the last day of the object location test, one of the two sample objects was moved to a different location.

### 4.10. Western Blot Analysis

Tissue was homogenized in radioimmunoprecipitation assay buffer (Cell Signaling, Danvers, MA, USA). Protein concentrations were determined using the Bio-Rad protein assay (Bio-Rad, Hercules, CA, USA). Western blot analyses were performed using 40 μg of protein. Briefly, samples were separated by 12% sodium dodecyl sulfate–polyacrylamide gel electrophoresis (SDS–PAGE) and transferred to polyvinylidene fluoride membranes (Merck Millipore, Burlington, MA, USA; 0.4 μm). The membranes were blocked in 5% bovine serum albumin (Bovogen Biologicals, Keilor East, Australia) in Tris-buffered saline and Tween-20 (Junsei Chemical, Tokyo, Japan) and incubated overnight at 4 °C with primary antibodies against APP/Aβ (6E10, Biolegend, San Diego, CA, USA; 1:1000), tau (HT7) (Thermo, Waltham, MA, USA; 1:1000), p-tau (AT8) (Thermo, Waltham, MA, USA; 1:1000), BDNF (Thermo, Waltham, MA, USA; 1:1000), TrkB (Thermo, Waltham, MA, USA; 1:1000), p-TrkB (Thermo, Waltham, MA, USA; 1:1000), CREB (Cell Signaling, Danvers, MA, USA; 1:1000), p-CREB (Cell Signaling, Danvers, MA, USA; 1:1000), and GAPDH (Cell Signaling, Danvers, MA, USA; 1:1000). After incubation with horseradish peroxidase-conjugated secondary antibodies (Sigma, Burlington, MA, USA; 1:5000) for 1 h at RT, immunodetection was performed using an enhanced chemiluminescence detection kit (GE Healthcare, Chicago, IL, USA). The protocol of Rosen et al. (60 μg of protein per well, 15% gel, and after transferred to membrane, the membrane incubated at 100 °C for 15 min with PBS) was used for the detection of Aβ multimers [78].

### 4.11. Statistical Analysis

SPSS 19.0 for Windows (SPSS Inc., Chicago, IL, USA) was used for the statistical analysis. The results were presented as mean ± standard error of the mean (SEM) values. Mean escape latency results for the Morris water maze test were analyzed using two-way repeated measures ANOVA, followed by Bonferroni’s test. The other data were analyzed using one-way ANOVA followed by Fisher’s LSD.

## Figures and Tables

**Figure 1 ijms-24-06827-f001:**
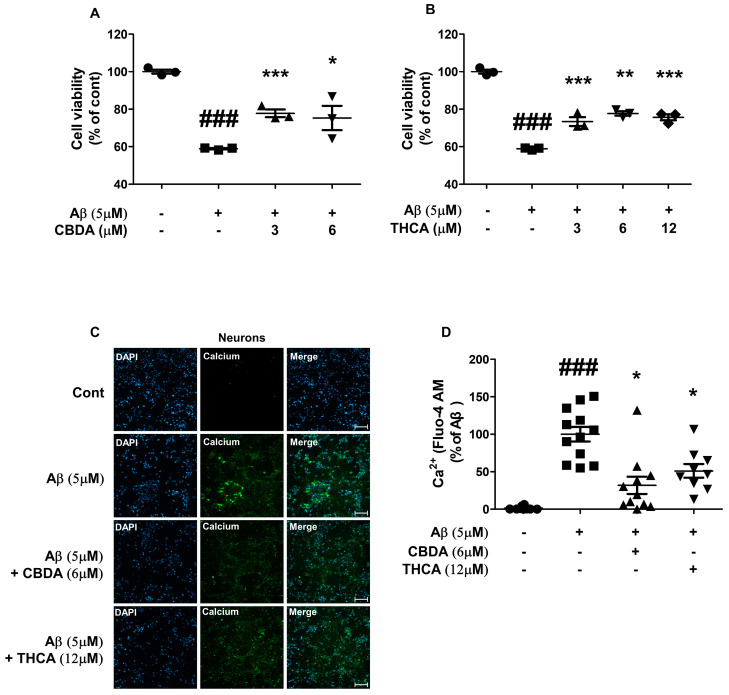
Effects of CBDA and THCA on neuroprotection and Ca^2+^ levels in primary neurons. Neuroprotective effect of (**A**) CBDA and (**B**) THCA. (**C**) Representative fluorescence images of Fluo-4 AM-positive Ca^2+^. Nuclei were stained using DAPI. Scale bar = 100 μm. (**D**) Fluorescence Ca^2+^ intensities. (mean ± SEM, ### *p* < 0.001 vs. PBS-treated, * *p* < 0.05, ** *p* < 0.005, and *** *p* < 0.001 vs. Aβ_1–42_-treated, one-way ANOVA followed by Fisher’s LSD; *n* = 3 per group).

**Figure 2 ijms-24-06827-f002:**
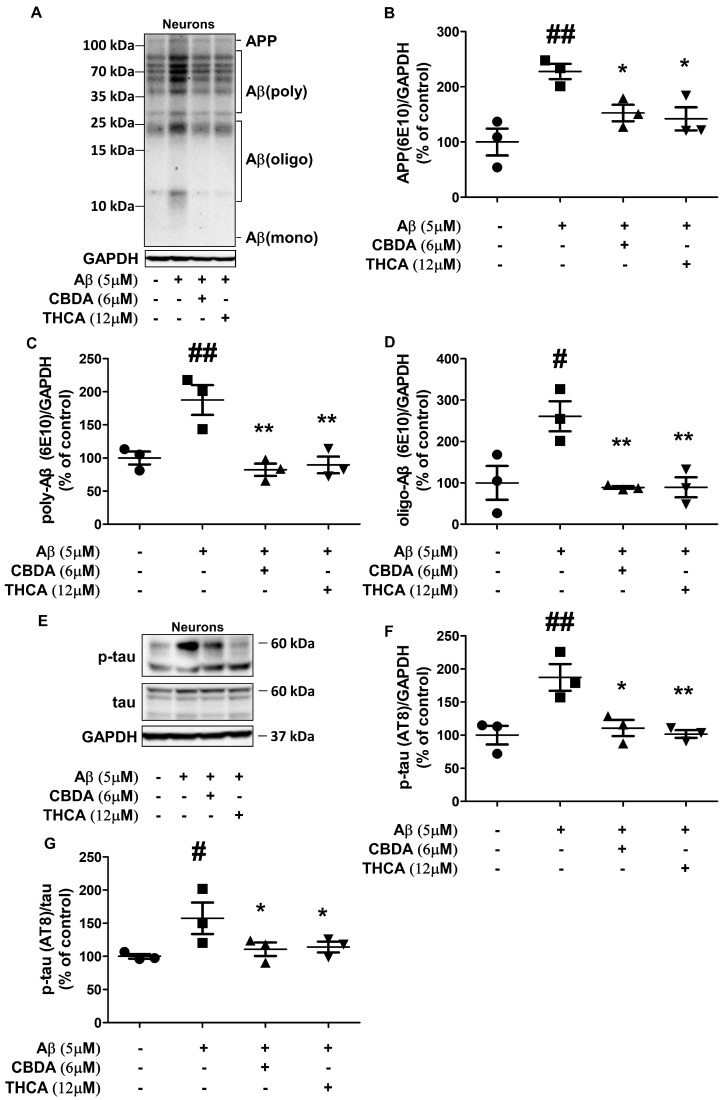
Effects of CBDA and THCA on Aβ and p-tau levels in primary neurons. (**A**) Representative Western blots of APP/Aβ proteins. (**B–D**) Western blot densitometry results for (**B**) APP, (**C**) polymeric Aβ, and (**D**) oligomeric Aβ. (**E**) Representative Western blots of tau and p-tau protein. (**F**,**G**) Western blot densitometry results for (**F**) p-tau (AT8), (**G**) p-tau (AT8)/tau. (mean ± SEM, # *p* < 0.05 and ## *p* < 0.005 vs. PBS-treated, * *p* < 0.05 and ** *p* < 0.005 vs. Aβ_1–42_-treated, one-way ANOVA followed by Fisher’s LSD; test; *n* = 3 per group).

**Figure 3 ijms-24-06827-f003:**
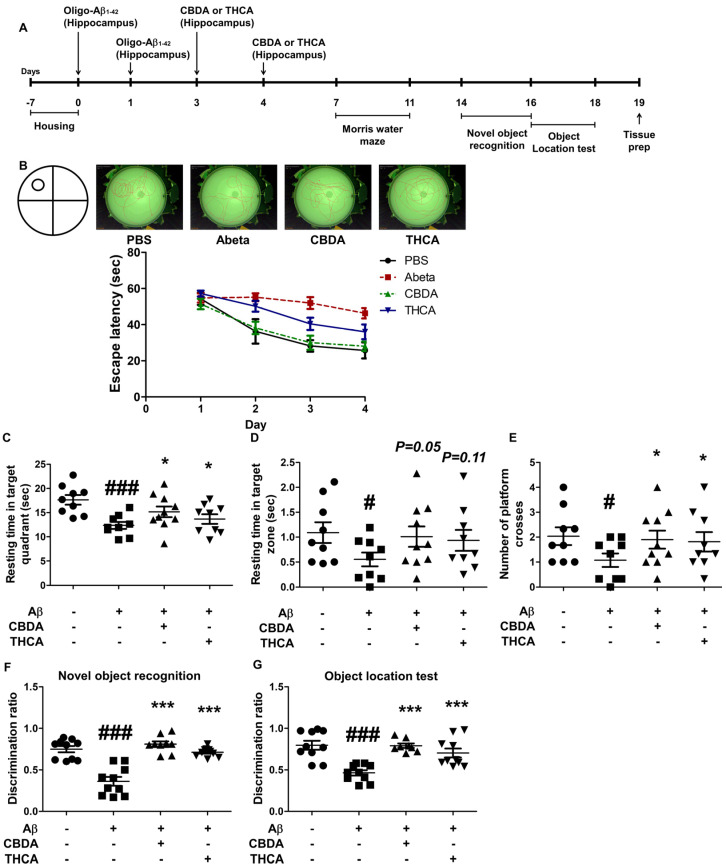
Effects of CBDA and THCA on learning and memory in Aβ_1–42_ -treated mice. (**A**) Schedule used for the behavioral tests. (**B**) Mean escape latency results for the Morris water maze test (mean ± SEM, two-way repeated measures ANOVA, followed by Bonferroni’s post-hoc test; *n* = 10 per group). Time spent in (**C**) target quadrant area and (**D**) platform area. (**E**) Number of crosses in the platform area. (**F**) Discrimination ratio results for the novel object recognition test. (**G**) Discrimination ratio results for the object location test. (mean ± SEM, # *p* < 0.05 and ### *p* < 0.001 vs. PBS-treated, * *p* < 0.05 and *** *p* < 0.001 vs. Aβ_1–42_-treated, one-way ANOVA followed by Fisher’s LSD; *n* = 10 per group).

**Figure 4 ijms-24-06827-f004:**
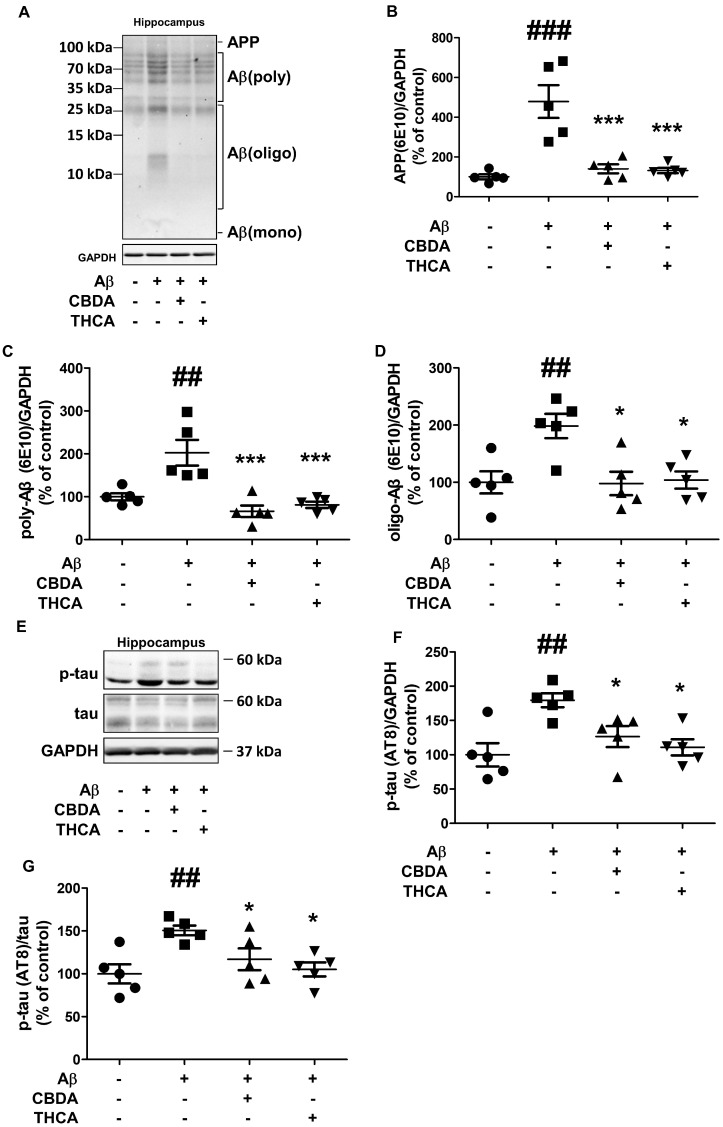
Effects of CBDA and THCA on hippocampal Aβ and p-tau levels in Aβ_1–42_-treated mice. (**A**) Representative Western blots of APP/Aβ proteins. (**B**–**D**) Western blot densitometry results for (**B**) APP, (**C**) polymeric Aβ, and (**D**) oligomeric Aβ. (**E**) Representative Western blots of tau and p-tau (AT8) protein. (**F**,**G**) Western blot densitometry results for (**F**) p-tau (AT8), (**G**) p-tau (AT8)/tau. (mean ± SEM, ## *p* < 0.005 and ### *p* < 0.001 vs. PBS-treated, * *p* < 0.05 and *** *p* < 0.001 vs. Aβ_1–42_-treated, one-way ANOVA followed by Fisher’s LSD; *n* = 5 per group).

**Figure 5 ijms-24-06827-f005:**
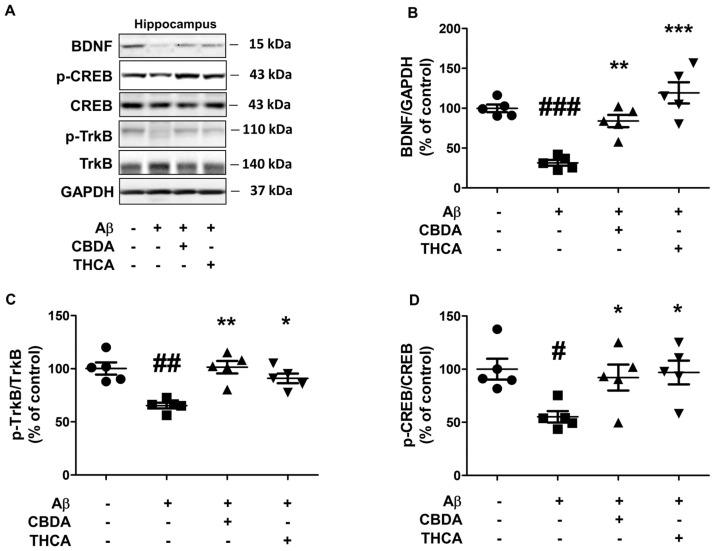
Effects of CBDA and THCA on hippocampal BDNF/CREB signaling-related protein levels in Aβ_1–42_-treated mice. (**A**) Representative Western blots of BDNF, p-CREB, and p-TrkB proteins. (**B**–**D**) Western blot densitometry results for (**B**) BDNF, (**C**) p-TrkB/TrkB, and (**D**) p-CREB/CREB. (mean ± SEM, # *p* < 0.05, ## *p* < 0.005, and ### *p* < 0.001 vs. PBS-treated, * *p* < 0.05, ** *p* < 0.005, and *** *p* < 0.001 vs. Aβ_1–42_-treated, one-way ANOVA followed by Fisher’s LSD; *n* = 5 per group).

## Data Availability

The data generated and analyzed during the current study are available from the corresponding author upon reasonable request.

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
