# Peer review of "The Cannabinoids, CBDA and THCA, Rescue Memory Deficits and Reduce Amyloid-Beta and Tau Pathology in an Alzheimer’s Disease-like Mouse Model"

_ijms, 2023, doi:10.3390/ijms24076827_

Round 1
Reviewer 1 Report
The work by Juyong Kim et al., investigated the protective effects the cannabinoids CBDA and THCA in cortical neurons in culture and in vivo in female mice following an acute treatment of Ab1-42 oligomers. The authors hypothesized that these cannabinoids “ameliorate AD by modulating Ca2 levels, hippocampal pathology and cognitive decline.” They reported that both compounds attenuated memory deficits, amyloid-beta and tau pathology through calcium-related BDNF/TRkB/CREB signaling. Based on these results, they conclude that both compounds may represent therapeutic agents for Alzheimer’s disease (AD).
The endogenous cannabinoid system has become as an important pharmacological target for treating neurodegenerative diseases, including AD (Fernández-Ruiz J et al., Handb Exp Pharmacol. 2015; 231: 233–259). While the focus has been on the neutral cannabinoids CBD and THC, new research indicates that cannabinoids in their acidic state display healing properties. The original idea of this study is to apply CBDA and THCA on in vitro and in vivo AD models. The study is well designed and many results are presented.
However, they are several points to consider:
General comments
1. This study was not conducted in patients with AD and did not use samples from AD patients. Therefore, it is incorrect to use the term AD alone when describing the results. Mice and rats do develop the disease; most cellular and animal models also do not recapitulate the disease in terms of pathology and symptoms. Using “AD-like models, AD-like features” or “cellular models of AD” “AD-related amyloid-beta pathology,” “AD-associated memory impairments” is more appropriate.
2. I understand that the authors chose to target intracellular Ca2+ because indeed it increased Ca2+ level has been shown in the brain of patients with AD (Calvo-Rodriguez, M. Nat. Commun. 2020, 11, 1–17), in several in vitro models of AD (Li L et al., Am. J. Chin. Med. 2010, 38, 113–125; Yallampalli S et al, Neurosci. Lett. 1998, 251, 105–108; Ekinci FJ et al., Mol. Brain Res. 2000, 76, 389–395) and to be functionally related to most pathological features and pathogenic factors of AD (Aβ plaque formation, tau hyperphosphorylation, apoptosis and synaptic dysfunction, etc.).
However, the mechanisms underlying the Ab1-42 oligomer-induced increase in Ca2+ levels are largely undescribed. Intracellular Ca2+ can result from Ca2+ influx from the extracellular space, Ca2+ leakage from intracellular Ca2+ stores, increased NMDAR-dependent Ca2+ influx (De felice FG et al., J. Biol. Chem. 2007, 282, 11590–11601).
More importantly, the authors did not show the causal relationship between the reversal of Ab1-42 oligomer-induced increase in Ca2+ levels in primary cortical neurons by CBDA and THCA involving TrkB/BDNF/CREB signaling and the decrease in amyloid and p-tau levels. For example, can TrkB antagonists counteract the cannabinoid effects on these parameters?
The situation could be even more complex knowing that Ca2+ accumulated in neuronal cells also has the ability to induce the formation of the endogenous Aβ oligomer, the aggregation of amyloid-beta and hyperphosphorylated tau (Guan et al., Int J Mol Sci 2021 31;22:5900; Itkin et al. PLoS ONE 2011, 6, e18250). Activation of cannabinoid receptor 2 is also known to protect rat hippocampal neurons against Aβ-induced neuronal toxicity (Zhao J et al., Neurosci Lett. 2020 Sep 14;735:135207). In summary, the authors should adapt their introduction and discussion accordingly.
3. Another concern is the statistical analysis. Surprisingly, at lines 345-436, the authors stated, “The results are presented as the mean ± standard error of the mean (SEM). All data were analyzed using a Student’s t-test”. In fact, in all figure legend we can read, “one way-ANOVA followed by the Student-t-test”. First, t-test tell us whether or not two groups have the same mean and cannot be used following ANOVA. ANOVA tells us which parameters are significant, but it does not explicitly tell which groups have means different from one another. To find out, we use a post-hoc test, such as Tukey’s multiple comparison test, Bonferonni or Fisher’s LSD test (the latter does not correct for multiple comparison).
Additionally, two-way ANOVA should have been used to compare the effects of treatment and time on the escape latency in the behavioral Morris water maze test.
A major revision should be made by the authors for the statistical analysis of all data. The description of the results is also too general. I recommend providing the fold or a % change between the groups, adding the exact p values for one-way ANOVA and post-hoc test.
4. The manuscript contains many inaccuracies in most parts of the manuscript. A typical example is “Calcium dyshomeostasis affects various signaling pathways in the nerve cells, results in damage, and eventually develops into severe AD [13, 14].
Providing more specific details will bring clarity, more precise understanding, and reproducibility of data, and this concerns most parts of the manuscript (title, introduction, results, material and methods). Some interpretations must be nuanced.
The images are too small; title on the Y-axis and the error bars (SEM) hard to read. I suggest changing histograms to dots to show individual data, mean and SEM, to clearly show the differences between the groups.
All Western blot images provided must have the molecular weight marker indicated and all lanes identified.
Specific comments
Major points
1- Title
The authors have observed the beneficial effects of CBDA and THCA specifically in primary cultures of cortical neurons and in mice following Ab1-42 oligomer treatment. Therefore, the title should clearly indicate that the study used AD-like models related to amyloid-beta induced pathology, especially since the effects of cannabinoids on tau phosphorylation were not investigated in a pure model of tauopathy or in 3xAD-Tg mice that display paired helicoidal filaments and neurofibrillary tangles over time. For example, the expression “during AD development” line 65 is inappropriate. Based on all these comments, the title has to be changed accordingly.
2-Introduction
A more informative descriptive of amyloid beta-related Ca2+ signaling pathways will help the non-expert in the field understand why the authors focused on BDNF/TrkB/ signaling over others (including PI3K/AKT, PLC/Inositol triphosphate). Additionally, it is unclear whether CBDA and THCA bind to G-protein-coupled CB1 or CB2 receptors (if so similarly or differently) and whether there is a link between the function of these receptors and regulation of intracellular Ca2+ signaling.
3- Results
a) Neuronal cell death
The effects of cannabinoids on the death of cortical neuronal cells treated with Ab1-42 oligomers should be further described by providing the factor or % of decrease in cell viability, the increase due to each cannabinoid as well as statistic values. Indeed, the reviewer is not convinced by “the significant” differences as presented in Figure 1 A and 1B.
Similar Efficacy is observed for both cannabinoids, but CBDA is more potent (3 µM) than THCA against Ab-induced neuronal death. Is it a matter of chemical structure or binding to CB receptors? Can treatment of CBDA or THCA alone without Ab-treatment regulate Ca2+ levels? This may be a good control to show.
b) Ca2+ levels in primary cultures of cortical neurons
The images in Figure 1C are very dark. Green fluorescence of Ca2+ Fluo-4aM and blue nuclei stained with DAPI can barely be seen. The brightness and contrast enhancement will help appreciate the ~50% decrease in intracellular Ca2+ levels in CBDA- or THCA-treated neurons as compared to the increased Ca2+ levels induced by Ab1-42 alone;
Is there a reason why the concentration of 3 µM of CBDA, which produced significantly higher effects in protecting against increased neuronal death induced by Ab1-42 oligomers, was not employed?
c) Amyloid b and p-tau levels in primary cultured cortical neurons
Neuronal cultures were treated with Ab1-42 oligomers or PBS (Control) for 24h. Figure 2A shows different amyloid-beta species that were recognized by 6E10 in all the four groups.
- Could the polymers be in fact oligomers of higher molecular size?
- The presence all Ab species in the control group as in treated groups is very surprising (Figure 2A, lane 1) and deserve an explanation.
- It is unclear how the ratios shown in Figure 2C, D was calculated. In the western blot image Fig 2, it would be informative to know which samples were run in lanes 5 to 10, 15 to 20 and 25-30.
d) Phospho-tau and tau levels
Western blot image Fig2-tau: The entire Western blot image must be provided in order to assess the p-tau/tau ratio. THCA appears to decrease tau levels in lanes 14 and 25.
Is the GAPDH image similar in Figure 1A and Figure 1E? If yes that means that the membrane of p-tau is available and should be provided.
The authors tested only the AT8 antibody. It would be instructive to mention the phosphorylated epitopes (Ser202/Thr205) in the text and graphics. It is known that tau can be phosphorylated in multiple sites and the authors did not test whether CBDA and THCA decreases tau phosphorylation at other epitopes (e.g., those recognized by AT180 (Thr231), AT270 (Thr181) etc.).
e) Memory performances in mice
- As mentioned in the introduction, all the statistical analysis incorrect. In particular, escape latency data should be analyzed by two-way ANOVA to learn how time and treatment (independent variables), in combination, regulate the escape latency (a dependent variable). The authors claimed significant differences, but they were not reported in the text (line 109). Data should be described in detail by providing the % of change between the groups with p value for ANOVA and for post-hoc test.
- The behavioral tests were carried out from day7 to day18 after Ab1-42 oligomers. During that period of time, oligomers may certainly have been transformed to multimers, protofibrils, and even fibrils. How can you be sure that CBDA and THCA act only on memory processes?
- Cannabinoids (e.g.,THC) could also interact directly with amyloid-b protofibrils and disrupt their structure (Kanchi & Dasmahapatra,
- Fig3 B: it is preferable to enlarge the graph and symbols and to change PBS, CBDA and THCA written alone without Ab to Ab+PBS, Ab+CBDA and Ab+THCA in Figure 1C.
f) Ab1-42 oligomers acute injection in the hippocampus on amyloid p-tau levels
- Did the authors verify 18 days later if the Aβ1–42 oligomers cleared or not, if the accumulation of amyloid species did not extend beyond the hippocampus or they believed that cannot happen?
- The authors did not show that neuronal cell death occurred in the hippocampus of mice treated with Ab1-42 oligomers and that it was restored by the two cannabinoids. Western blot analysis with NeuN antibody could be performed.
- The problem of statistics applies here too
e) TrkB/BDNF/CREB signaling pathways
- The statistical analysis must be corrected.
- The entire Western blot image of p-TrkB must be shown to check its concordance with that of TrkB. Same requirements for BDNF and GAPDH. Each lane has to be labeled and the molecular weight marker indicated.
4- Discussion
Unfortunately, the discussion took up most ideas set out the introduction without criticism. The results were not discussed in relation to published findings on the effects of cannabinoids on neuronal TrkB, differences from neural CBD and THC for their actions on calcium channels, reduction of amyloid and memory performance in models of AD, etc. A major revision of the discussion is expected.
Minor points
- The complete names of CBDA and THCA are not given
- Lines 15, 50, 193: CBDA and THCA cannot be “precursor metabolites” of CBD and THC. In fact, they are acidic variants of the neutral CBD and THC, respectively, and their precursors.
Can CBDA and THCA be converted to the respective neutral forms in neurons or in vivo? For example, GABA is formed from the decarboxylation of glutamic acid (Glu), and both GABA and Glu are important biologically active food components.
- Line 27 “Taken together, CBDA and THCA may represent useful therapeutic agents for preventing or treating AD.” In the present study, the authors did not show that cannabinoids prevented the toxic effects of Ab1-42 oligomers since they applied after the amyloid-beta treatment.
- For consistency, primary cultures of hippocampal neurons could have been prepared. For Fig1 title, I suggest removing neuroprotection and write, “…. on cell death (or cell viability) and Ca2+ levels in primary cultures of cortical neurons (or cortical neuronal primary cultures)
- Was the Aβ1–42 oligomer preparation checked, for instance by transmission electron microscopy? In which specific area of the hippocampus were Ab1-42 oligomers injected? The best control would have been scrambled Aβ1–42.
- Ca2 fluorescence intensity in neurons: total or mean intensity in the graphs?
- *** p<0.001 vs Ab treated is missing in the legend of Figure 1.
- Figure 2 legend (F): specify p-tau/GAPDH
- Line 105 ..learning and memory performances (not memory processes) are evaluated
- Line 341: the protocol of Rosen et al. should be briefly explained. Twice the word “protocol” in this sentence
- English language is sometimes imperfect and the manuscript should be checked and corrected.
Line 33: “ More than 100 years have passed since Alois Alzheimer’s was reported” is unclear
Line 42: “For dementia patients” has to be replaced with “For demented patients.”
Line 278 “Aβ1–42 Stereotaxic surgery”: I suggest, “Intrahippocampal stereotaxic injection of Ab1-42

Author Response
Reviewer #1
General comments
- This study was not conducted in patients with AD and did not use samples from AD patients. Therefore, it is incorrect to use the term AD alone when describing the results. Mice and rats do develop the disease; most cellular and animal models also do not recapitulate the disease in terms of pathology and symptoms. Using “AD-like models, AD-like features” or “cellular models of AD” “AD-related amyloid-beta pathology,” “AD-associated memory impairments” is more appropriate. : Thank you for your comments. We revised the term as you advised accordingly. 2. We understand that the authors chose to target intracellular Ca2+ because indeed it increased Ca2+ level has been shown in the brain of patients with AD (Calvo-Rodriguez, M. Nat. Commun. 2020, 11, 1–17), in several in vitro models of AD (Li L et al., Am. J. Chin. Med. 2010, 38, 113–125; Yallampalli S et al, Neurosci. Lett. 1998, 251, 105–108; Ekinci FJ et al., Mol. Brain Res. 2000, 76, 389–395) and to be functionally related to most pathological features and pathogenic factors of AD (Aβ plaque formation, tau hyperphosphorylation, apoptosis and synaptic dysfunction, etc.). However, the mechanisms underlying the Ab1-42 oligomer-induced increase in Ca2+ levels are largely undescribed. Intracellular Ca2+ can result from Ca2+ influx from the extracellular space, Ca2+ leakage from intracellular Ca2+ stores, increased NMDAR-dependent Ca2+ influx (De felice FG et al., J. Biol. Chem. 2007, 282, 11590–11601). More importantly, the authors did not show the causal relationship between the reversal of Ab1-42 oligomer-induced increase in Ca2+ levels in primary cortical neurons by CBDA and THCA involving TrkB/BDNF/CREB signaling and the decrease in amyloid and p-tau levels. For example, can TrkB antagonists counteract the cannabinoid effects on these parameters? : Thank you for your comments about the causal relationship of amyloid beta oligomerization with calcium levels. As CBDA and THCA is not well described about regulation of signaling pathways, we focused on how amyloid beta increased calcium levels in the system. As you mentioned, exact mechanims how CBDA and THCA inhibited the rise in calcium concentration is studying. The situation could be even more complex knowing that Ca2+ accumulated in neuronal cells also has the ability to induce the formation of the endogenous Aβ oligomer, the aggregation of amyloid-beta and hyperphosphorylated tau (Guan et al., Int J Mol Sci 2021 31;22:5900; Itkin et al. PLoS ONE 2011, 6, e18250). Activation of cannabinoid receptor 2 is also known to protect rat hippocampal neurons against Aβ-induced neuronal toxicity (Zhao J et al., Neurosci Lett. 2020 Sep 14;735:135207). In summary, the authors should adapt their introduction and discussion accordingly. : Thank you for your commnets. we added the documents you mentioned in the introduction and discussion section. 3. Another concern is the statistical analysis. Surprisingly, at lines 345-436, the authors stated, “The results are presented as the mean ± standard error of the mean (SEM). All data were analyzed using a Student’s t-test”. In fact, in all figure legend we can read, “one way-ANOVA followed by the Student-t-test”. First, t-test tell us whether or not two groups have the same mean and cannot be used following ANOVA. ANOVA tells us which parameters are significant, but it does not explicitly tell which groups have means different from one another. To find out, we use a post-hoc test, such as Tukey’s multiple comparison test, Bonferonni or Fisher’s LSD test (the latter does not correct for multiple comparison). : Thank you for your kind review, we additionally conducted post hoc analysis with Fisher’s LSD. Additionally, two-way ANOVA should have been used to compare the effects of treatment and time on the escape latency in the behavioral Morris water maze test. : Thank you for your kind review, Two-way ANOVA analysis of mean escape latency (i.e., the time required to locate the escape platform) in the Morris water maze test revealed statistically significant between four groups (p=0.04). A major revision should be made by the authors for the statistical analysis of all data. The description of the results is also too general. I recommend providing the fold or a % change between the groups, adding the exact p values for one-way ANOVA and post-hoc test. : Thank you for your kind review. We described the % change between the groups and p-value in the result part of the paper. 4. The manuscript contains many inaccuracies in most parts of the manuscript. A typical example is “Calcium dyshomeostasis affects various signaling pathways in the nerve cells, results in damage, and eventually develops into severe AD [13, 14]. Providing more specific details will bring clarity, more precise understanding, and reproducibility of data, and this concerns most parts of the manuscript (title, introduction, results, material and methods). Some interpretations must be nuanced. : Thank you for your kind review. We modified that sentence and the language has been modified through the company. We have attached a certificate.. The images are too small; title on the Y-axis and the error bars (SEM) hard to read. I suggest changing histograms to dots to show individual data, mean and SEM, to clearly show the differences between the groups. All Western blot images provided must have the molecular weight marker indicated and all lanes identified. : Thank you for your kind review. We modified band and graph format. Specific comments Major points 1- Title The authors have observed the beneficial effects of CBDA and THCA specifically in primary cultures of cortical neurons and in mice following Ab1-42 oligomer treatment. Therefore, the title should clearly indicate that the study used AD-like models related to amyloid-beta induced pathology, especially since the effects of cannabinoids on tau phosphorylation were not investigated in a pure model of tauopathy or in 3xAD-Tg mice that display paired helicoidal filaments and neurofibrillary tangles over time. For example, the expression “during AD development” line 65 is inappropriate. Based on all these comments, the title has to be changed accordingly. : Thank you for your kind review. We completely agree with your opinion. Therefore, We revised the title and the sentence you mentioned. 2-Introduction A more informative descriptive of amyloid beta-related Ca2+ signaling pathways will help the non-expert in the field understand why the authors focused on BDNF/TrkB/ signaling over others (including PI3K/AKT, PLC/Inositol triphosphate). Additionally, it is unclear whether CBDA and THCA bind to G-protein-coupled CB1 or CB2 receptors (if so similarly or differently) and whether there is a link between the function of these receptors and regulation of intracellular Ca2+ signaling. : Thank you for your kind review. The effect of BDNF in Alzheimer's disease and its correlation are further described in discussion part. When CB is activated, the calcium concentration in the cell decreases. It is also clear that CBDA and THCA act as agonists to CB. But there is no clear prior literature yet whether CBDA and THCA bind to CB directly. 3- Results a) Neuronal cell death The effects of cannabinoids on the death of cortical neuronal cells treated with Ab1-42 oligomers should be further described by providing the factor or % of decrease in cell viability, the increase due to each cannabinoid as well as statistic values. Indeed, the reviewer is not convinced by “the significant” differences as presented in Figure 1 A and 1B. Similar Efficacy is observed for both cannabinoids, but CBDA is more potent (3 µM) than THCA against A-induced neuronal death. Is it a matter of chemical structure or binding to CB receptors? Can treatment of CBDA or THCA alone without Ab-treatment regulate Ca2+ levels? This may be a good control to show. : Thank you for your kind review. We described the % change between the groups and p-value in the result part of the paper. We do not have data on why CBDA exhibits a greater neuronal protection effect than THCA. In addition, even when referring to previous studies, there is no literature comparing the differences between CBDA and THCA on neuroprotective effects, so it is hard to answer accurately. Previous literature suggests that both CBDA and THCA reduce calcium concentrations (Faouzi, Wakano et al. 2022). In normal neuronal cells, calcium concentration is about 1.37% in in vitro. Therefore, we assume that even CBDA or THCA treatment affecting calcium concentration, it would be hard to compare the decreased concentration with 1.37%. We are currently planning to alleviate Alzheimer's disease using CBDA and THCA through oral administration in a chronic AD-like model. As you advised, We will also check the effects of CBDA and THCA alone. b) Ca2+ levels in primary cultures of cortical neurons The images in Figure 1C are very dark. Green fluorescence of Ca2+ Fluo-4aM and blue nuclei stained with DAPI can barely be seen. The brightness and contrast enhancement will help appreciate the 50% decrease in intracellular Ca2+ levels in CBDA- or THCA-treated neurons as compared to the increased Ca2+ levels induced by Ab1-42 alone; Is there a reason why the concentration of 3 µM of CBDA, which produced significantly higher effects in protecting against increased neuronal death induced by A1-42 oligomers, was not employed? : Thank you for your kind review. As you said, we increased the resolution and size of calcium images. The reason why we employee 6 µM fo CBDA was the neuroprotective effect was greater than 3 µM and there was no significant statistical difference between 3 µM and 6 µM of CBDA. c) Amyloid b and p-tau levels in primary cultured cortical neurons Neuronal cultures were treated with A1-42 oligomers or PBS (Control) for 24h. Figure 2A shows different amyloid-beta species that were recognized by 6E10 in all the four groups. - Could the polymers be in fact oligomers of higher molecular size? : Thank you for your kind review. According to other studies, the criteria for the molecular weight of oligo-amyloid beta are not clear. One literature covers up to 70 kDa as oligomers (Yang, Li et al. 2017). Since there is no clear boundary between oligomers and polymers, we quantified up to 5 polymers (25kDa) as oligomers. - The presence all Ab species in the control group as in treated groups is very surprising (Figure 2A, lane 1) and deserve an explanation. : Thank you for your kind review. It is not clear why amyloid beta is found in the same pattern without treating oligo-amyloid beta. Previous literatures also show that amyloid beta is detected in the same pattern without treatment oligo-amyloid beta (Sondag, Dhawan et al. 2009) and in transgenic mouse model (Vale, Alonso et al. 2010). - It is unclear how the ratios shown in Figure 2C, D was calculated. In the western blot image Fig 2, it would be informative to know which samples were run in lanes 5 to 10, 15 to 20 and 25-30. : Thank you for your kind review. We think it's the same opinion as you said above. Since there is no clear boundary between oligomers and polymers, polymer and oligomer were divided and quantified Based on 25kDa. Samples in lanes 5 to 10, 15 to 20 and 25-30 are other experiment that we run together with CBDA and THCA treated samples. Those are not related with this manuscript. d) Phospho-tau and tau levels Western blot image Fig2-tau: The entire Western blot image must be provided in order to assess the p-tau/tau ratio. THCA appears to decrease tau levels in lanes 14 and 25. Is the GAPDH image similar in Figure 1A and Figure 1E? If yes that means that the membrane of p-tau is available and should be provided. : Thank you for your kind review. In order to measure various protein expressions in one membrane, we cut the membrane and used based on the molecular weight of the target proteins. The authors tested only the AT8 antibody. It would be instructive to mention the phosphorylated epitopes (Ser202/Thr205) in the text and graphics. It is known that tau can be phosphorylated in multiple sites and the authors did not test whether CBDA and THCA decreases tau phosphorylation at other epitopes (e.g., those recognized by AT180 (Thr231), AT270 (Thr181) etc.). : Thank you for your kind review. We described the p-tau's epitope information (AT8) in the text and graphics. As you said, we will use various p-tau antibodies in further experiments. The reason for choosing only the AT8 form in various phosphorylation forms of tau proteins is that only the AT8 phosphorylation form was changed in the amyloid beta-injected animal model (Brouillette, Caillierez et al. 2012). e) Memory performances in mice - As mentioned in the introduction, all the statistical analysis incorrect. In particular, escape latency data should be analyzed by two-way ANOVA to learn how time and treatment (independent variables), in combination, regulate the escape latency (a dependent variable). The authors claimed significant differences, but they were not reported in the text (line 109). Data should be described in detail by providing the % of change between the groups with p value for ANOVA and for post-hoc test. : Thank you for your kind review, Two-way ANOVA analysis of mean escape latency (i.e., the time required to locate the escape platform) in the Morris water maze test revealed statistically significant between four groups (p=0.04) and described p-value in the result part of the paper. - The behavioral tests were carried out from day7 to day18 after Ab1-42 oligomers. During that period of time, oligomers may certainly have been transformed to multimers, protofibrils, and even fibrils. How can you be sure that CBDA and THCA act only on memory processes? - Cannabinoids (e.g.,THC) could also interact directly with amyloid-b protofibrils and disrupt their structure (Kanchi & Dasmahapatra, J Mol Graph Model 2021, 105: 107889). Whether THCA or CBDA could act in the same way can be discussed. : Thank you for your kind review. In this study, we investigated the mitigating effects of CBDA and THCA on Alzheimer's disease. We know that CBDA and THCA treatment effect on detailed and exact mechanisms are under investigation including our group. Therefore, we chose the calcium concentration and its regulation after CBDA or THCA treatment. High concentrations of calcium are known to promote the aggregation and production of amyloid beta and the pathogenesis of tau protein. As you mentioned, previous studies have shown that cananabinoids inhibit the aggregation and accumulation of amyloid beta. In particular, there is a prior literature that CBD and THC inhibit the aggregation and accumulation of amyloid beta (Janefjord, Mååg et al. 2014). However there was no prior literature on the impact of CBDA and THCA on amyloid beta aggregation. Although we did not attach the data, It was confirmed that CBDA and THCA have better neuroanl cell protection effects than CBD and THC. - Fig3 B: it is preferable to enlarge the graph and symbols and to change PBS, CBDA and THCA written alone without A to A+PBS, A+CBDA and A+THCA in Figure 1C. : Thank you for your kind review. We revised it. f) A1-42 oligomers acute injection in the hippocampus on amyloid p-tau levels - Did the authors verify 18 days later if the Aβ1–42 oligomers cleared or not, if the accumulation of amyloid species did not extend beyond the hippocampus or they believed that cannot happen? : Thank you for your kind review. Previous literature also shows that amyloid beta, injected into the hippocampus, affects the memory-related brain region, cortex(Faucher, Mons et al. 2016). Therefore, it is expected to affect not only the hippocampus but also other brain areas. In AD patients, hippocampus is a major part to affect severity of disease development (Rao, Ganaraja et al. 2022). - The authors did not show that neuronal cell death occurred in the hippocampus of mice treated with Ab1-42 oligomers and that it was restored by the two cannabinoids. Western blot analysis with NeuN antibody could be performed. : Thank you for your kind review. We confirm that CBDA and THCA inhibit apoptosis by amyloid beta in nerve cells. Although it was desirable to check directly in animal models, we confirmed changes in the BDNF/CREB signaling pathway. Since the BDNF/CREB signal directly affects apoptosis, the expression of apoptosis proteins is also expected to change. We are currently planning to alleviate Alzheimer's disease using CBDA and THCA through oral administration in a chronic AD-like model. As you advised, We will also check the effects of CBDA and THCA on apoptosis. - The problem of statistics applies here too : Thank you for your kind review, we additionally conducted post hoc analysis with Fisher’s LSD. e) TrkB/BDNF/CREB signaling pathways - The statistical analysis must be corrected. : Thank you for your kind review, we additionally conducted post hoc analysis with Fisher’s LSD. - The entire Western blot image of p-TrkB must be shown to check its concordance with that of TrkB. Same requirements for BDNF and GAPDH. Each lane has to be labeled and the molecular weight marker indicated. : Thank you for your kind review. In order to measure various protein expressions in one membrane, a membrane was cut and used based on the molecular weight of the target proteins. 4- Discussion Unfortunately, the discussion took up most ideas set out the introduction without criticism. The results were not discussed in relation to published findings on the effects of cannabinoids on neuronal TrkB, differences from neural CBD and THC for their actions on calcium channels, reduction of amyloid and memory performance in models of AD, etc. A major revision of the discussion is expected. : Thank you for your kind review. We have supplemented the points you mentioned above in the introduction and discussion part. Minor points - The complete names of CBDA and THCA are not given : Thank you for your kind review. We explained full names of each compounds in abstract and introduction parts. - Lines 15, 50, 193: CBDA and THCA cannot be “precursor metabolites” of CBD and THC. In fact, they are acidic variants of the neutral CBD and THC, respectively, and their precursors. : Thank you for your kind review. We changed precursor metabolites to acidic variants. Can CBDA and THCA be converted to the respective neutral forms in neurons or in vivo? For example, GABA is formed from the decarboxylation of glutamic acid (Glu), and both GABA and Glu are important biologically active food components. : Thank you for your kind review. CBDA and THCA are not expected to change to CBD and THC in the body. This is because CBDA and THCA require very high temperatures to change to CBD and THC (Seo, Jeong et al. 2022), and in the PK study, CBDA and THCA reach the brain when oral administrated (Anderson, Low et al. 2019) . - Line 27 “Taken together, CBDA and THCA may represent useful therapeutic agents for preventing or treating AD.” In the present study, the authors did not show that cannabinoids prevented the toxic effects of A1-42 oligomers since they applied after the amyloid-beta treatment. : Thank you for your kind review. We deleted the word about preventive efficacy. - For consistency, primary cultures of hippocampal neurons could have been prepared. For Fig1 title, I suggest removing neuroprotection and write, “…. on cell death (or cell viability) and Ca2+ levels in primary cultures of cortical neurons (or cortical neuronal primary cultures) : Thank you for your kind review. We changed fig 1 title into « CBDA and THCA treatment decreases cell death and Ca2+ levels in primary cultures of cortical neurons » - Was the Aβ1–42 oligomer preparation checked, for instance by transmission electron microscopy? In which specific area of the hippocampus were A1-42 oligomers injected? The best control would have been scrambled Aβ1–42. : Thank you for your kind review. We did not confirmed the degree of aggregagtion of amyloid-beta. However, amyloid beta was prepared in the same way to introduce intracellular calcium influx as described in (Islam, Cho et al. 2022). In addition, several experimental results showed similar degree of apoptosis level with different preparation of amyloid beta. Therefore, we assumed that amyloid-beta we used could induces changes in calcium concentration and apoptosis. We injected into ML = 1.30 mm, AP = -2.00 mm, and DV = -2.20 mm based on Bregma, and this area was injected into the dentate gyrus (DG) site. - Ca2 fluorescence intensity in neurons: total or mean intensity in the graphs? : Thank you for your kind review. Regions of interest were 422,500 μm2 and randomly selected from each well. It was measured 3-4 areas per well. The Image J analysis program measured the entire fluo-4 AM fluorescence signal intensity. - *** p<0.001 vs Ab treated is missing in the legend of Figure 1. : Thank you for your kind review. We revised it. - Figure 2 legend (F): specify p-tau/GAPDH : Thank you for your kind review. We revised it. - Line 105 ..learning and memory performances (not memory processes) are evaluated : Thank you for your kind review. We measured the cognitive behavioral experiments NOR and OLT. It was thought to be an experimental method to measure the memory of objects and spaces, so the word ‘memory’ was used. - Line 341: the protocol of Rosen et al. should be briefly explained. Twice the word “protocol” in this sentence : Thank you for your kind review. We revised it. - English language is sometimes imperfect and the manuscript should be checked and corrected. : Thank you for your kind review. We modified that sentence and the language has been modified through the company. We have attached a certificate. Line 33: “ More than 100 years have passed since Alois Alzheimer’s was reported” is unclear : Thank you for your kind review. We revised the sentence to ‘It's been more than 100 years since AD was first reported. But there is currently no ade-quate treatment for AD.’ Line 42: “For dementia patients” has to be replaced with “For demented patients.” : Thank you for your kind review. We revised it. Line 278 “Aβ1–42 Stereotaxic surgery”: I suggest, “Intrahippocampal stereotaxic injection of A1-42 : Thank you for your kind review. We revised it.

Reviewer 2 Report
The manuscript by Kim et al uses acid cannabinoids CBDA and THCA to rescue alterations in Aβ1-42 treated primary neurons and in an Aβ1-42 treated mice. They find these cannabinoids partially improve the alterations induced by Aβ1-42. However, there are several points that should be improved or amended.
Major points:
1. The statistics have not been done correctly. If an ANOVA analysis is done, then pairwise comparisons must be done with a post hoc analysis. Student's t test should not be used for this, as it is only useful for comparing means when there are only two data series. Therefore, the statistics must be redone for all figures, and since the significance in many cases is small, some comparisons may not be significant after recalculation.
2. The Calcium experiments are poorly designed and provide no real information. First, fluo-4 is loaded for an unusually long period of 24h, which is probably toxic per se. The usual is 1-2h at most. Apparently, these live fluo-4 loaded cells are then stained with DAPI to stain nuclei, not to see viability. But DAPI does not readily cross cell membranes, resulting in minimal staining with DAPI. And finally a single measurement of fluo-4 fluorescence is made as representative of the calcium level. This assumes no understanding of how Ca2+ signaling works in the cell. Ca2+ is a second messenger that has continuously fluctuating levels in the cytosol (and in various organelles). Fluorescent dyes such as fluo-4 allow continuous measurement of Ca2+ variations over time at the single cell level, and only in that way can information about changes in the Ca2+ signal be obtained. The single measurement that has been taken can only give information about the number of dead cells with permanently elevated Ca2+ levels. The images shown in Fig. 1 should be withdrawn or replaced.
3. The quality of the figures in general is very poor, they have very low resolution and the sizes of the western blot images, bar graphs and text fonts are all too small and impossible to read.
Minor point.
In the introduction, it should be explained what CBDA and THCA are and their full names should be given at least once.
Author Response
Reviewer #2
Major points:
- The statistics have not been done correctly. If an ANOVA analysis is done, then pairwise comparisons must be done with a post hoc analysis. Student's t test should not be used for this, as it is only useful for comparing means when there are only two data series. Therefore, the statistics must be redone for all figures, and since the significance in many cases is small, some comparisons may not be significant after recalculation.
: Thank you for your kind review, we additionally conducted post hoc analysis with Fisher’s LSD.
- The Calcium experiments are poorly designed and provide no real information. First, fluo-4 is loaded for an unusually long period of 24h, which is probably toxic per se. The usual is 1-2h at most. Apparently, these live fluo-4 loaded cells are then stained with DAPI to stain nuclei, not to see viability. But DAPI does not readily cross cell membranes, resulting in minimal staining with DAPI. And finally a single measurement of fluo-4 fluorescence is made as representative of the calcium level. This assumes no understanding of how Ca2+ signaling works in the cell. Ca2+ is a second messenger that has continuously fluctuating levels in the cytosol (and in various organelles). Fluorescent dyes such as fluo-4 allow continuous measurement of Ca2+ variations over time at the single cell level, and only in that way can information about changes in the Ca2+ signal be obtained. The single measurement that has been taken can only give information about the number of dead cells with permanently elevated Ca2+ levels. The images shown in Fig. 1 should be withdrawn or replaced.
: Thank you for your kind review. fluo-4 dyeing material was treated for 1 hour. The purpose of this experiment is to determine whether CBDA and THCA decreases the calcium concentration raised by the amyloid beta. As you said, calcium is a secondary messenger, calcium levels constantly changes. However, since calcium concentration is increased by amyloid beta, what we want to show through this data was CBDA and THCA have calcium concentration inhibitory effects against amyloid beta. Calcium concentrations can be measured high due to the dead cells, but to exclude such effects as much as possible, we washed them three times with PBS after stain. We described more details about how to measure calcium concentration.
- The quality of the figures in general is very poor, they have very low resolution and the sizes of the western blot images, bar graphs and text fonts are all too small and impossible to read.
: Thank you for your kind review. We modified band and graph format.
Minor point.
In the introduction, it should be explained what CBDA and THCA are and their full names should be given at least once.
: Thank you for your kind review. We explained full names of each compounds in abstract and introduction parts.

Reviewer 3 Report
In the present study by Kim et al., the authors investigated the effects of CBDA and THCA that are contained in hemp extracts on amyloid beta toxicity. Given the recent circumstances that medical and/or recreational use of hemp have been legalized in many states/provinces in the USA and other countries, the topic of this study is of importance in the field of neurodegeneration and of social significance. However, the current manuscript is unacceptable for several reasons. The authors used the 6E10 anti-amyloid beta antibody and stated that CBDA/THCA treatment decreased APP in mouse cortical neurons and mouse brains. This is unacceptable because 6E10 does not recognize mouse APP. They used “oligomeric” Abeta without confirming formation of Abeta oligomers or citing appropriate refs. They performed multiple comparisons without correcting errors, i.e., repeating t-test for comparison of more than three groups. Multiple comparison such as Tucky or Bonferroni test must be performed. They stated that CBDA/THCA treatment decreased Abeta levels in mouse neurons and brains without demonstrating no mechanisms. The link between Abeta treatment and reduced p-Tau by CBDA/THCA has not been provided. All in all, the manuscript does not meet the criteria for publication of this journal.
Author Response
Reviewer #3
In the present study by Kim et al., the authors investigated the effects of CBDA and THCA that are contained in hemp extracts on amyloid beta toxicity. Given the recent circumstances that medical and/or recreational use of hemp have been legalized in many states/provinces in the USA and other countries, the topic of this study is of importance in the field of neurodegeneration and of social significance. However, the current manuscript is unacceptable for several reasons.
The authors used the 6E10 anti-amyloid beta antibody and stated that CBDA/THCA treatment decreased APP in mouse cortical neurons and mouse brains. This is unacceptable because 6E10 does not recognize mouse APP.
: Thank you for your kind review. It is not clear why amyloid beta is found in the same pattern without treating oligo-amyloid beta. Previous literature also shows that amyloid beta is detected in the same pattern without treatment oligo-amyloid beta (Sondag, Dhawan et al. 2009) and transgenic mouse model (Vale, Alonso et al. 2010).
They used “oligomeric” Abeta without confirming formation of Abeta oligomers or citing appropriate refs.
: Thank you for your kind review. We did not confirmed the degree of aggregagtion of amyloid-beta. However, amyloid beta was prepared in the same way to introduce intracellular calcium influx as described in (Islam, Cho et al. 2022). In addition, several experimental results showed similar degree of apoptosis level with different preparation of amyloid beta. Therefore, we assumed that amyloid-beta we used could induces changes in calcium concentration and apoptosis.
They performed multiple comparisons without correcting errors, i.e., repeating t-test for comparison of more than three groups. Multiple comparison such as Tucky or Bonferroni test must be performed. They stated that CBDA/THCA treatment decreased Abeta levels in mouse neurons and brains without demonstrating no mechanisms. The link between Abeta treatment and reduced p-Tau by CBDA/THCA has not been provided.
: Thank you for your kind review, we additionally conducted post hoc analysis with Fisher’s LSD. In this study, we investigated the mitigating effects of CBDA and THCA on Alzheimer's disease. However, we also know that CBDA and THCA lack an accurate mechanism for inhibiting the expression of amyloid beta and pathogenesis of tau protein. We focused on calcium among various mechanisms. High concentrations of calcium are known to promote the aggregation and production of amyloid beta and the pathogenesis of tau protein.
All in all, the manuscript does not meet the criteria for publication of this journal.

Reviewer 4 Report
The authors find that CBDA and THCA which extracted from hemp can protect primary neurons and mice from Aβ1-42 treatment, the results are interesting.
I only have a few minor comments:
1. Figure 3b, the label of graph is not clear, please use different colors or the other ways to distinguish the four groups.
2. There are some pare is not consistent in introduction and discussion:
a. Authors said “ CBDA and THCA function directly in the brain because of their ability to cross the blood-brain barrier “, while in discussion “CBDA can directly affect the brain because of its ability to penetrate the BBB [26]. CBDA showed anti-hyperalgesia, anti-inflammation, anti-nausea, anti-anxiety, and anti-seizure effects in animal models Although THCA has poor BBB penetration ability, it exhibits anti-convulsant, anti-inflammation, and anti-nausea effects”. And according to the references, the description in discussion is reasonable.
b. It seems that CBDA and THCA was reported to have anti-convulsant effects.
Author Response
Reviewer #4
- Figure 3b, the label of graph is not clear, please use different colors or the other ways to distinguish the four groups.
: Thank you for your kind review. We modified graph format.
- There are some pare is not consistent in introduction and discussion:
- Authors said “ CBDA and THCA function directly in the brain because of their ability to cross the blood-brain barrier “, while in discussion “CBDA can directly affect the brain because of its ability to penetrate the BBB [26]. CBDA showed anti-hyperalgesia, anti-inflammation, anti-nausea, anti-anxiety, and anti-seizure effects in animal models Although THCA has poor BBB penetration ability, it exhibits anti-convulsant, anti-inflammation, and anti-nausea effects”. And according to the references, the description in discussion is reasonable.
: Thank you for your kind review. Both CBDA and THCA show BBB permeability. THCA is less BBB permeability than CBDA, so we described separately. We revised the contents about this.
- It seems that CBDA and THCA was reported to have anti-convulsant effects.
: Thank you for your kind review. We added information on the anti-convulsive effect of CBDA.

Round 2
Author Response
Reviewer 1
1- Statistical analysis
There seems to be some confusion in the statistical analysis. The authors stated:
Line 399: “All the data were analyzed using the Fisher’s LSD.”
Lines 124-125: “Two-way ANOVA analysis of mean escape latency (i.e., the time required to locate the escape platform) in the Morris water maze test revealed statistically significant between four groups (p=0.04)
Legend of Figures 1-5: one-way ANOVA followed by Fisher’s LSD.
Therefore, I wonder if the authors are familiar with statistics.
The description of the results should start by stating that the “One-way ANOVA results in a difference between treatments that is significant (p value) or not significant (p-value);
If ANOVA is significant, a pairwise comparison is done with a post hoc test (p-values with 3 digits after the period or expressed as to E-notation when the exponent cannot be conveniently displayed).
: Thank you for your comments. We modified statistical analysis method as « Mean escape latency results for the Morris water maze test were analyzed using two-way repeated measures ANOVA, followed by Bonferroni’s test. The other data were analyzed using the Fisher’s LSD. » in lane 323-332
The most problematic is the two-way ANOVA analysis. The authors did not perform it contrary to what was stated in lines 124, 125, “Two-way ANOVA analysis of mean escape latency (i.e., the time required to locate the escape platform) in the Morris water maze test revealed statistically significant between four groups (p=0.04)“,
The authors should show how treatment and time (2 factors) affect the escape latency (response variable) and whether or not there is an interaction effect between the factors on the response variable. Therefore, they must show the p-value for the interaction between treatment and time, the p-value for treatment, and the p-value for time.
: Thank you for your comments. As you said, we marked each p values as « interaction (p=0.04), Treatment (p=0.00), Time (p=0.00)) » in lane 126-129
2- Discussion
The discussion remains poor. There is no detailed critical comparison with the published literature.
Repetitions
Lane 223 “And its receptor, the cannabinoid receptor, performs various functions in the nervous system”
Lane 224: …”various studies have been conducted on the function of cannabinoid receptors in the central nervous system. C”
Lane 228: “cannabinoid receptors are expressed in various parts of the brain and perform many functions [55].”
: Thank you for your comments. We revised and supplemented the contents in discussion section. In lane 227-238, the sentences modified as « And its receptor, the cannabinoid receptor, performs various functions in the nervous system. Cannabinoid receptors are expressed in astrocytes, microglia, and neurons [56, 57]. And cannabinoid receptors are involved in the survival of nerve cells, synaptic plasticity and development of dendrites [58-60]. Cannabinoid receptors are expressed not only in the peripheral nervous system but also in the central nervous system [61]. So, it affects various neurodegenerative diseases, including Parkinson's disease and Alzheimer's disease [62, 63]. Therefore, various studies have been conducted on the function of cannabinoid receptors in the central nervous system. In particular, many studies have been conducted on neurodegenerative diseases using CBD and THC, representative agonists of cannabinoid receptors [64, 65]. CBD and THC have potential therapeutic efficacy in clinical trials for Parkinson's disease [66, 67]. Accordingly, the development of drugs targeting cannabinoid receptors is progressing [68]. »
Lack of precision and mistakes
Line 35: there is no “adequate”; in fact, there is currently no treatment for AD.
: Thank you for your comments. Lane 37, we deleted the word ‘adequate’
Line 53: the activation of CB: do the authors meant CB receptor 51 or 2)?
: Thank you for your comments. Lane 55, It means both CB1 and CB2. So we change the sentence as « the activation of CB1 and CB2 inhibits N-methyl-D-aspartate (NMDA) receptors and lowers calcium concentrations, showing neuroprotective effects [15, 16]. »
Lines 208-209 “The FDA introduces BDNF as an evaluation marker for the efficacy of AD treatments [45]. In fact, ref 45 deals with biomarkers for Alzheimer’s disease. BDNF was mentioned for MRI measures of hippocampal volume not at all for the efficacy of AD treatments
: Thank you for your comments. Lane 212, we modified that sentance ; « BDNF is attracted as an potential evaluation marker for the efficacy of AD treatments »
Line 210 … BDNF overexpression or its injection into AD animal models demonstrated therapeutic efficacy for AD [46]: which parameters: symptoms, neuropathology?
: Thank you for your comments. Overexpression of BDNF showed improvement in cognitive behavioral experiments and affected synaptic development. When BDNF was injected into the brain, it showed improvement in cognitive behavioral experiments.
3- Graphical
The size of the symbols particular the star (*) must be increased, as well the symbols – and + under each graph.
Figure 1C is still dark. DAPI is not visible.
: Thank you for your kind review. We modified brightness of Fig 1C and increased size of symbols.
4-English language
Still need improvement
Title: in an Alzheimer’s disease-like mouse model
Line 207: “AND”, at the beginning of a sentence
Line 393 "incubated at 100°C for 15 min with PBS after transferred to membrane "... replace by “and then transferred to the membrane”
Line 253 modulating Ca2+ homeostasis fundamental to neuronal viability and function.... replace by …homeostasis which is fundamental
: Thank you for your kind review. We completely agree with your opinion. Therefore, We revised the title and the sentence you mentioned.
Line 350: “Regions of interest that were 422 500 μm2 were randomly selected from each well respectively and measured 3-4 areas per well”. This sentence is unclear; What “respectively” refers to?
: Thank you for your comments. The meaning of the word 'respectively' means three to four pictures taken in every one well.
5- Micellaneous
Lines 208-209 “The FDA introduces BDNF as an evaluation marker for the efficacy of AD treatments [45]”. In fact, ref 45 deals with biomarkers for Alzheimer’s disease. BDNF was mentioned for MRI measures of hippocampal volume not for the efficacy of AD treatments
: Thank you for your comments. Lane 212, we modified that sentance ; « BDNF is attracted as an potential evaluation marker for the efficacy of AD treatments »

Reviewer 2 Report
The revised version of the manuscript has solved only in part the points I had raised.
1. Statistics is now made correctly with post-hoc analysis, but most of the effects found have a quite low level of significance due to high variability.
2. The explanations on the calcium and DAPI experiments are not satisfactory. The fluo-4 loading period is changed from 24h to 1h without explanation. Was that an error? No mention is made on the use of DAPI in live cells. And washing with PBS is not useful to change the high Ca2+ level inside dead cells. Those experiments could at most measure cell viability.
3. All the figures have been changed from bar plots to box plots, but the resolution, sizes of text fonts o significance symbols are still impossible to read.
4. The images of the GAPDH controls in the western blots of Fig. 2 and 4 have been substituted by new different ones. Also, some of the western blots in Fig. 4 look different from the ones in the original version. I can´t see any explanation for these changes.
In summary, I believe that several important questions remain open about this work, which require much further elaboration by the authors.
Author Response
Reviewer 2
- Statistics is now made correctly with post-hoc analysis, but most of the effects found have a quite low level of significance due to high variability.
: Thank you for your kind review. We think it's because there weren't many samples of animal testing (n=10). We are currently planning to alleviate Alzheimer's disease using CBDA and THCA through oral administration in a chronic AD-like model. As you advised, In order to reduce this high variability, we will increase the number of animals.
- The explanations on the calcium and DAPI experiments are not satisfactory. The fluo-4 loading period is changed from 24h to 1h without explanation. Was that an error?
: Thank you for your kind review. There was confusion in the experimental method. During fluo-4 dyeing, tissue slides were dyed for 24 hours, and cells were dyed for 1 hour. Based on what you said, we modified the experimental method based on the research note. I'm sorry to confuse you. In addition, we carefully checked whether there were any other errors in the experiment method.
No mention is made on the use of DAPI in live cells. And washing with PBS is not useful to change the high Ca2+ level inside dead cells. Those experiments could at most measure cell viability.
: Thank you for your kind review. As you said, in order to see the effect of controlling calcium concentration, it is most accurate to see the change in calcium concentration over time in a single cell. However, in Figure 1A and B, it was confirmed that CBDA and THCA inhibit neuronal apoptosis caused by amyloid beta. Previous literature suggests that amyloid beta increases calcium concentration, and increased calcium concentration induce apoptosis (Ekinci, Linsley et al. 2000) (Itkin, Dupres et al. 2011). And the dead cell debris cause inflammation(Boada-Romero, Martinez et al. 2020). In Figure 3c, we wanted to see if CBDA and THCA inhibit the overall calcium concentration increased by amyloid beta. Therefore, the overall calcium concentration was measured without excluding dead cells.
- All the figures have been changed from bar plots to box plots, but the resolution, sizes of text fonts o significance symbols are still impossible to read.
: Thank you for your kind review. We modified brightness of Fig 1C and increased size of symbols.
- The images of the GAPDH controls in the western blots of Fig. 2 and 4 have been substituted by new different ones. Also, some of the western blots in Fig. 4 look different from the ones in the original version. I can´t see any explanation for these changes.
: Thank you for your kind review. Original submission and revised version of GAPDH images were different, but from same experiment set with different mice. Therefore, you may feel the difference. However, GAPDH expression was not changed which means that loading was equally performed to check the different expression levels of each protein. As we submitted in previous whole western blotting, we marked only one out of different experimental sets.
Therefore, there is no change in tendency because it is a detected image in the same group sample.

Reviewer 3 Report
I have no further comment.
Author Response
We appreciate for your time and effort on the manuscript.
Round 3
Reviewer 1 Report
This version of the manuscript has been considerably improved by the authors, both in terms of figures and statistics
There are still a few minor points to correct:
1- p-value: it is preferablebetter to put 3 or more digits after the decimal point in cases where p<0.00 is written to have a precise information of the p-value.
For example, in lines 82, 83,8487, 126,127,131, 139, 140 etc. The entire text must be verified.
2- line 134 p-value=0.11 is not significant, line 136 p-value =0.25 is also not significant. The authors probably wrote the values wrong and should correct them.
3- Line 408: I think” the other data were analyzed using one-way ANOVA followed by the Fisher’s LSD test not just by Fisher’s LSD

Author Response
Reviewer 1.
1- p-value: it is preferable better to put 3 or more digits after the decimal point in cases where p<0.00 is written to have a precise information of the p-value.
For example, in lines 82, 83,84 87, 126,127,131, 139, 140 etc. The entire text must be verified.
: Thank you for your comments. We marked all the p-values up to the third digit below the decimal point. As one-way ANOVA followed by Fisher’s LSD statistical method, the p-value is expressed up to three decimal point. Therefore, if the p-value is 0.000, it is expressed as p<0.001. The modifications are as follows.
- Line 83-86: Neuronal cell death was markedly increased in primary neurons treated with Aβ1-42 (5 μM) by 70% (p<0.001), whereas CBDA (3 μM (75% (p<0.001)) and 6 μM (78% (p=0.009))) and THCA (6 (79% (p=0.004)) and 12 μM (79% (p<0.001))) significantly suppressed neuronal cell death (Fig. 1A, B).
- Line 87-91 : The fluorescence intensity of Ca2+ was significantly increased in Aβ1-42-treated neurons compared with PBS-treated neurons (100% (p<0.001)); however, this increase was significantly ameliorated by 6 μM CBDA (32% (p=0.005)) and 12 μM THCA (51% (p=0.010)) treatment (Fig. 1C, D).
- Line 127-130 : Two-way ANOVA analysis of mean escape latency (i.e., the time required to locate the escape platform) in the Morris water maze test revealed statistically significant between four groups (interaction (p=0.047), Treatment (p<0.0001), Time (p<0.0001))
- Line 134-147 : On day 5 of the probe test, Aβ1-42-treated mice remained in the target quadrant (p<0.001) and platform area (p=0.035) for a significantly shorter time compared with mice treated with PBS. For Aβ1-42-treated mice, CBDA or THCA treatment resulted in a longer time in the target quadrant (CBDA; p=0.008, THCA; p=0.030) and platform area (CBDA; p=0.059, THCA; p=0.114) (Fig. 3C, D). The number of times crossing the platform area was significantly reduced in Aβ1-42-treated mice compared with the PBS-treated mice (p=0.025). CBDA (p=0.026) or THCA (p=0.044) treatment resulted in an increase in the number of crossings in Aβ1-42-treated mice (Fig. 3E). During the novel object phase, mice treated with Aβ1-42 + CBDA (p<0.001) or THCA (p<0.001) spent more time exploring the novel object and exhibited significantly higher discrimination ratios compared with Aβ1-42-treated mice (Fig. 3F). In the object location test, mice treated with Aβ1-42 + CBDA (p<0.001) or THCA (p<0.001) also spent more time examining the displaced object and exhibited sig-nificantly higher discrimination ratios compared with Aβ1-42-treated mice (Fig. 3G).
In addition, the graph is marked with 1, 2, and 3 digits according to the p-value so that readers can know the statistical significance by just looking at the graph. In addition, the legend of each figure is marked as follows.
- figure 1. , ###p < 0.001 vs. PBS-treated, *p < 0.05, **p < 0.005 and ***p < 0.001 vs. Aβ1-42-treated, one-way ANOVA followed by Fisher’s LSD
- figure 2. , #p < 0.05 and ##p < 0.005 vs. PBS-treated, *p < 0.05 and **p < 0.005 vs. Aβ1-42-treated, one-way ANOVA followed by Fisher’s LSD
- figure 3. , #p < 0.05 and ###p < 0.001 vs. PBS-treated, *p < 0.05 and ***p < 0.001 vs. Aβ1-42-treated, one-way ANOVA followed by Fisher’s LSD
- figure 4. ##p < 0.005 and ###p < 0.001 vs. PBS-treated, *p < 0.05 and ***p < 0.001 vs. Aβ1-42-treated, one-way ANOVA followed by Fisher’s LSD
- figure 5. #p < 0.05, ##p < 0.005 and ###p < 0.001 vs. PBS-treated, *p < 0.05, **p < 0.005 and ***p < 0.001 vs. Aβ1-42-treated, one-way ANOVA followed by Fisher’s LSD
2- line 134 p-value=0.11 is not significant, line 136 p-value =0.25 is also not significant. The authors probably wrote the values wrong and should correct them.
: Thank you for your comments. The p- value in line 136 has been modified to 0.025.
In the Morris water maze experiment, the time spent in the target zone where the platform is located was not statistically increased in the CBDA and THCA treatment groups compared to the amyloid beta treatment groups.
For CBDA, the p value was 0.059, and for THCA, the p value was 0.114. Therefore, we wrote that the CBDA and THCA treatment groups stayed longer than the amyloid beta treatment group.
The information is as follows.
- Line 136- 138 : For Aβ1-42-treated mice, CBDA or THCA treatment resulted in a longer time in the target quadrant (CBDA; p=0.008, THCA; p=0.030) and platform area (CBDA; p=0.059, THCA; p=0.114)
3- Line 408: I think” the other data were analyzed using one-way ANOVA followed by the Fisher’s LSD test not just by Fisher’s LSD
: Thank you for your comments. We modified statistical analysis method as « Mean escape latency results for the Morris water maze test were analyzed using two-way repeated measures ANOVA, followed by Bonferroni’s test. The other data were analyzed using one-way ANOVA followed by the Fisher’s LSD.

Reviewer 2 Report
The paper has been improved in this last version, although I still think the level of significance of most findings is low and the calcium and DAPI experiments have technical problems and do not reveal much more than cell viability.
Author Response
Reviewer 2.
The paper has been improved in this last version, although I still think the level of significance of most findings is low and the calcium and DAPI experiments have technical problems and do not reveal much more than cell viability.
: Thank you for your comments. we were able to improve on the parts we missed. The goal of the calcium concentration measurement test was to focus on apoptosis, so we could not reflect the parts you advised me before. we will consider what you advised in further experiments.
